# Fast-Gated 16 × 1 SPAD Array for Non-Line-of-Sight Imaging Applications

**Marco Renna [1] , Ji Hyun Nam [2] , Mauro Buttafava [1] , Federica Villa [1] , Andreas Velten [2] and Alberto Tosi [1],***

[1] Dipartimento di Elettronica, Informazione e Bioingegneria (DEIB), Politecnico di Milano, I-20133 Milan, Italy; marco.renna@polimi.it (M.R.); mauro.buttafava@polimi.it (M.B.); federica.villa@polimi.it (F.V.)

[2] Computational Optics Group, University of Wisconsin—Madison, Madison, WI 53706, USA; jnam26@wisc.edu (J.H.N.); velten@wisc.edu (A.V.)

* Correspondence: alberto.tosi@polimi.it; Tel.: +39-02-2399-6174

**Abstract:** In this paper we present a novel single-photon detector specifically designed for Non-Line-Of-Sight (NLOS) imaging applications within the framework of the DARPA REVEAL program. The instrument is based on a linear 16 × 1 Complementary Metal-Oxide-Semiconductor (CMOS) Single-Photon Avalanche Diode (SPAD) array operated in fast-gated mode by a novel fast-gating Active Quenching Circuit (AQC) array, which enables the detectors with sub-ns transitions thanks to a SPAD-dummy approach. The detector exhibits a timing resolution better than 50 ps (Full Width at Half Maximum - FWHM) at a measurement repetition rate up to 40 MHz, and provides 16 independent outputs compatible with commercial Time-Correlated Single-Photon Counting (TCSPC) instrumentation. The instrument has been experimentally characterized and operated in preliminary NLOS imaging acquisitions where a 40 × 60 cm hidden object is successfully reconstructed by scanning over a grid of 150 × 150 positions.

**Keywords:** SPAD array; fast-gated; Non-Line-Of-Sight (NLOS); single-photon; Time-Of-Flight (TOF)

## 1. Introduction

Optimal performance achieved throughout previous years by fast-gated SPAD detectors has fostered their diffusion in a wide range of different applications, from fluorescence lifetime measurements to time-resolved Near-InfraRed Spectroscopy (NIRS), quantum cryptography and Light Detection and Ranging (LIDAR) measurements based on Time-Of-Flight (TOF) techniques [1–6]. In particular, fast-gated single-photon detectors adopted in TOF measurements may allow to recover information on objects hidden from the direct Line-Of-Sight (LOS) of the imaging system, thanks to the acquisition of photons after multiple reflections through the scene, implementing the so-called Non-Line-Of-Sight (NLOS) imaging [7].

In a TOF measurement, the light reaching the detector includes contributions of both direct light (i.e., light travelling back and forth from the laser source to objects in the system LOS) and indirect light, which interacts with hidden parts of the scene before being detected after multiple reflections (commonly called bounces). Nevertheless, direct (or ballistic) photons generally outnumber indirect (or nonballistic) photons, making it difficult to discriminate multiple bounce contributions among the many detection events. Most TOF-based imaging techniques aim at removing these multiple bounce contributions, as they increase the background noise in acquiring the LOS image. Conversely, NLOS techniques rely on the detection of nonballistic photons to effectively detect objects hidden from the field of view of the imaging system. In order to effectively discriminate between ballistic and nonballistic photons, NLOS imaging systems make use of time modulated laser sources (either pulsed or amplitude modulated) and time-resolved single-photon detectors.

NLOS imaging techniques and systems gained interest as they enable the remote viewing of hidden scenarios dangerous or difficult to access, as NLOS imaging may be adopted in rescuing scenarios or hazardous industrial environments, in the clinical field to improve endoscopic analysis and surgeries and lastly in security and automotive applications. Nevertheless, all these scenarios require different operating parameters in terms of reconstruction volume (from a few $cm^3$ for endoscopic analysis up to tens of $m^3$ for room reconstructions) and light intensity, thus a NLOS imaging system should offer great flexibility and reconfigurability in order to satisfy different application areas.

In this paper, we present the design of a complete single-photon detector based on a novel 1D array of Complementary Metal-Oxide-Semiconductor (CMOS) Single-Photon Avalanche Diodes (SPADs) driven by an array of Active Quenching Circuits (AQCs) in fast-gated regime, developed within the framework of the DARPA REVEAL (Revolutionary Enhancement of Visibility by Exploiting Active Light-fields) program. The adoption of an integrated AQC allows for system scalability towards multipixel systems with no compromise on detector performance and strong cost reduction with respect to circuits based on discrete components. The system features 16 independent SPADs with the possibility to increase the number of detectors up to 64 pixels, still guaranteeing a time-gated operation with OFF/ON transitions < 1 ns, and an overall Single-Photon Timing Resolution (SPTR) lower than 50 ps (FWHM).

## 2. Non-Line-of-Sight Imaging

The use of NLOS imaging systems outside laboratories has been discouraged by high cost and bulkiness of imaging solutions, or otherwise by poor performance in terms of timing resolution and Signal-to-Noise Ratio (SNR). Nevertheless, NLOS imaging techniques have been applied both at radio and visible wavelengths. Systems operating at radio wavelengths have been developed and effectively applied to obtain low-resolution NLOS images through walls [8] and, taking advantage of mirror-like reflections, around-the-corner imaging [9] and motion detection [10]. At visible wavelengths most systems instead rely on coded controllable light sources, such as a projector or a laser source coupled with a galvo mirror, to illuminate the scene (and the hidden scene) [11], or on specular reflections in a window pane [12]. Photon TOF information is typically used in gated viewing and LIDAR applications [13], but can be applied also to detect nonballistic photons for NLOS imaging, and have been effectively used for transient imaging of multiple-bounce incoherent photons by means of streak cameras [14] or low cost Photonic Mixer Devices (PMDs) [15,16], which can be furtherly exploited to reconstruct scenes hidden from the field of view of the imaging system [17].

The first instrument to demonstrate the possibility to use TOF acquisition for NLOS imaging was based on a streak camera [18,19], providing 2 ps timing resolution and 1 cm lateral spatial resolution in the reconstruction. Nevertheless, the high cost, bulk and fragility of a streak camera discourage its use outside a research laboratory environment. Systems based on modulated light sources and PMDs were developed to overcome streak cameras limitations, thanks to PMDs' compactness and low cost. Nevertheless, these systems provide a timing resolution in the order of few nanoseconds and a spatial resolution of about three meters. Intensified Charge-Coupled Devices (I-CCDs) were also adopted in combination with a pulsed laser source, thus implementing the so-called laser gated viewing technique [20]. Despite portability and sub-nanosecond timing resolution, I-CCDs still exhibit scalability issues due to high costs, thus preventing their adoption in systems tailored to mass-market application.

Both streak cameras and I-CCD cameras allow for NLOS imaging but due to the limited number of photons acquired during every measurement cycle and the low measurement repetition rate, a complete reconstruction requires a long time. Given the high losses in the propagation path, it is mandatory to maximize the sensitivity, the Dynamic Range (DR) and the SNR of the NLOS system in order to reduce the overall acquisition time. These needs can be fulfilled by SPAD detectors, which guarantee high detection rate and picosecond timing resolution, besides low cost and compactness. NLOS tracking of an object position in an empty space with the use of a 32 × 32 SPAD array was demonstrated [21],

by using as scattering and imaging surface the floor, thus removing the limitation of a relay wall in front of the acquisition system.

The measurement DR can be furtherly increased by time-gating the SPAD detector in order to suppress the ballistic photon contribution, which is generally orders of magnitude greater than third bounce contribution, thus avoiding detector saturation and dramatically improving system performance in the reconstruction of the hidden scene [7].

In [7], the system is based on a single-pixel detector and a scanned pulsed laser source, and aims at full reconstruction of a NLOS scene with about 10 cm spatial resolution by using common surface materials and low light illumination, still guaranteeing reduced costs and complexity with respect to other NLOS systems, with no worsening of imaging performance. The detector here adopted is a single-pixel fast-gated SPAD featuring 20 μm active area diameter, SPTR < 30 ps (FWHM) and OFF/ON transitions < 150 ps (20–80%) [22,23], whereas the pulsed laser source is at 515 nm and provides sub-ps optical pulses at 55 MHz repetition rate with 50 mW average optical power. To evaluate improvements introduced by the adoption of a fast-gated SPAD detector, it can be noticed that laser optical power is an order of magnitude lower than what used in [18]. The detector is focused on a 1 cm$^2$ area and the laser beam scans a $1 \times 0.8$ m$^2$ area on the relay wall, leading to a 185 points dataset. The overall reconstruction times ranges between five minutes for a 1 s integration time per scanned point and 32 minutes for a 10 s integration time.

## 2.1. NLOS Systems and Reconstruction Techniques: State-of-the-Art

Recently, O'Toole et al. [24], proposed a NLOS reconstruction method based on a Light Cone Transform (LCT). Lindell et al. [25] demonstrated a reconstruction algorithm, f-k Migration, which is based on seismic imaging techniques. Both algorithms are capable of reconstructing complex NLOS scenes. However, they both use confocal measurement scheme where the illumination laser and the detector have to be focused on the same spot and scanned together. Therefore, it is naturally limited to single-pixel SPAD. Existing commercial SPAD arrays have been used for tracking of targets. The systems lack gating and have poor timing resolution and fill factor. Reconstructions of a similar quality to the ones created with single pixel SPADs have not been demonstrated. Chan et al. [26], successfully showed long range target-position retrieval algorithm for a single target in the hidden scene. They used single pixel SPAD and successively scanned four different positions on the visible wall to mimic SPAD array with four pixels, while the laser illumination position is fixed to one location. However, the method is also limited only to tracking a target. Prior SPAD-based reconstruction methods that do not rely on confocal capture schemes are filtered back-projection [7], and phasor-field [27] reconstructions. In this paper, we demonstrate the possibility of using a SPAD array for NLOS reconstruction using a non-confocal data acquisition scheme combined with a phasor-field method [27].

## 2.2. Improvements in NLOS Reconstruction: The Phasor-Field Method

The reconstruction method exploited in [7] is based on a filtered model presented in [18]. Typical computation times of few hours are required due to the large amount of data and the complex algorithms employed for the reconstruction [28]. The work presented in [28] reduces the computational resources (by up to three orders of magnitude) required to perform the reconstruction as it decreases the overall number of computation by considering only the data resulting from the intersects of several ellipsoids whose poles are the light source and the detector. Additionally, thanks to a novel phasor-field approach, NLOS imaging scenarios can be treated by taking advantage of well-established LOS imaging techniques, improving accuracy and robustness of NLOS reconstruction in complex scenarios [27]. By exploiting the techniques presented in [28] and [27], the previous technique [18] and other NLOS imaging methods can be outperformed it terms of both computation time and accuracy.

Given the advancements in reconstruction methods and techniques, an acquisition time of several minutes prevents the NLOS imaging of moving objects. In order to reduce the measurement acquisition time two main strategies can be adopted: (i) improve the measurement SNR, by increasing laser power

(but this leads to eye safety issues), or improving photon harvesting capability of the detector (larger SPAD active area or lenses with larger aperture); (ii) a multipoint detection strategy, by adopting 1D or 2D SPAD arrays, for detecting photons at multiple points on the relay wall in order to reduce the number of needed laser points and the total acquisition time [7].

## 3. Instrument Design

In the following sections, the architectures of both the fast-gating AQC array and the CMOS SPAD array are introduced, together with a complete 16 × 1 fast gated SPAD array specifically designed for NLOS imaging. A two-chip structure was adopted, with SPADs and AQCs fabricated in two different technologies: 0.35 μm HV CMOS by Fraunhofer IMS and 0.35 μm SiGe-BiCMOS by AMS (Austrian Micro Systems AG), in order to obtain the best performance for each device with a compromise on system integration. The 0.35 μm HV CMOS by Fraunhofer IMS allows the production of low-noise SPADs, whereas the 0.35 μm SiGe-BiCMOS process guarantees fast, low-jitter transistors for SPAD driving and signal readout.

### 3.1. Fast-Gating AQC Array

The newly developed fast-gating AQC is based on the architecture of the SPAD driver and quenching circuit presented in [29], featuring few minor modifications mainly tailored to reduce the overall number of external connections of the chip, allowing to design an array structure. The main features of the previous single-pixel SPAD driver and quenching circuit have been preserved: (i) a differential sensing of a SPAD–dummy pair allows the rejection of the gate coupling while keeping fast SPAD transients; (ii) a fast differential comparator, which guarantees avalanche current detection with very low jitter; (iii) a programmable hold-off logic circuit, needed to keep the SPAD OFF for a certain amount of time after every photodetection event. The main differences with respect to the previous fast-gating AQC ASIC are discussed below.

Considering the architecture of the SPAD pulser (i.e., the circuit driving the SPAD above/below breakdown voltage with sub-ns transitions, shown in Figure 1a), the only difference is the adoption of two Metal-Oxide-Semiconductor (MOS) transistors driven in ohmic regime (see $M_1$ and $M_2$ in Figure 1a) to modulate the resistance at emitter node of the reset BJT, thus allowing the variation of the peak current drawn from the SPAD anode node with an impact on the time needed to drive the SPAD from OFF to ON. The different dimensioning of M1 (W/L = 140 μm/0.35 μm) and M2 (W/L = 30 μm/0.35 μm), and the possibility to switch them ON/OFF independently, allow for three different operating conditions to adjust SPAD pulser transient times. For the experimental measurements here reported in the following, both M1 and M2 are always ON in order to have the shortest turn-on transition among the three different possibilities.

In order to reduce the overall number of connections, in this design the SPI communication of the previous version was replaced by a custom One Wire (OW) interface, which allows the programing of the chip by using a single connection.

Another modification aimed at the reduction of the chip external connections regards the readout thresholds of the fast SiGe front-end comparator. An internal band-gap reference (followed by a level-shifting stage) is introduced in this design, allowing the derivation of an internal reference which can be selected between two nominal values, 0.9 and 1.5 V. The internal voltage reference is connected to the SPAD side of the front-end comparator, thus only one external reference, connected to the dummy side, is required for setting comparator threshold. In the presented system, the 1.5 V internal voltage reference was chosen as it proved better timing performance with respect to 0.9 V.

The last difference with respect to previous AQC ASIC is the removal of the clock input for the hold off circuit: in this design, the 16-bit hold-off counter is clocked by either an internal ring-oscillator, which generates a nominal 100 MHz reference or by the external gate signal. When the internal clock is used, the hold-off time can be adjusted between 50 ns and 655 μs at 10 ns steps. When the external gate signal is used, the minimum hold-off time is such that the detector can be re-enabled at the successive

valid gate period after hold-off time, whereas the maximum hold-off time is equal to 216 gate periods. In the presented detection system, the gate signal is adopted as clock source of the hold-off counter, in order to reduce power consumption of the chip and to avoid the strong dependence of the internal ring-oscillator output frequency on operating temperature.

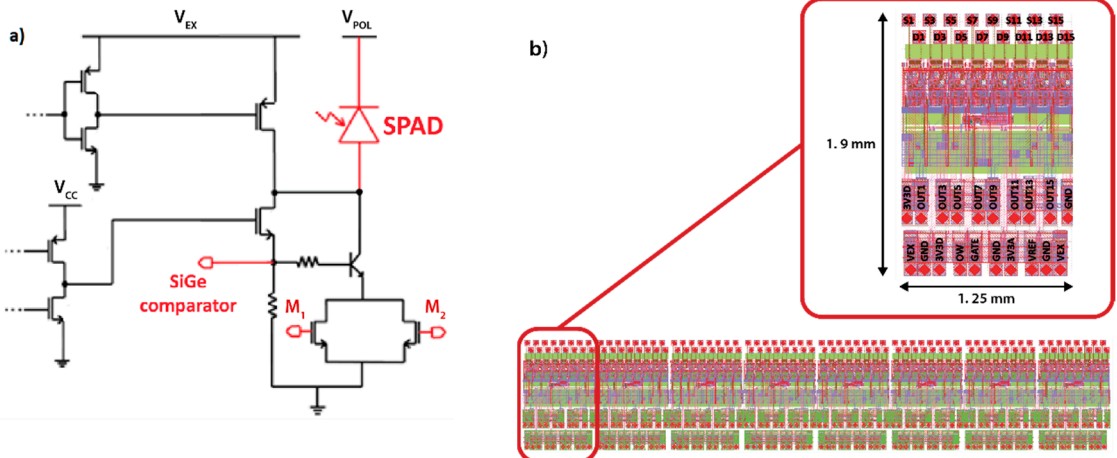

**Figure 1.** (**a**) Simplified schematic of the Single-Photon Avalanche Diode (SPAD) pulser circuit: the Bipolar Junction Transistor (BJT) transistor connected at SPAD anode node is active during the OFF-to-ON transient of the SPAD. The Metal-Oxide-Semiconductor (MOS) transistors M1 and M2, driven in ohmic regime, allow the variation of the current drawn by the BJT, thus the transient of the SPAD anode. An identic circuit (not shown) drives the dummy device and allows for differential sensing. (**b**) Layout of the complete $64 \times 1$ fast-gating Active Quenching Circuit (AQC) array and, in the inset, a single $8 \times 1$ block, composed of eight independent AQCs. On the top side are placed the contacts for the SPAD-dummy pairs, while on the bottom side two rows of bonding pads are placed for external connections.

The final $64 \times 1$ AQC array is made of eight independent blocks, each one composed by eight fast-gating AQCs (Figure 1b). Each block shares power supplies, configuration register, gate signal and external threshold voltage, while providing eight independent outputs (CMOS logic levels). The output pulse synchronous to a photon detection has a fixed 4 ns temporal duration, nevertheless the output buffer is not designed to drive 50 $\Omega$ impedance coaxial cables, and thus an external buffer is required. Each block requires three different power supplies: the voltage rail for the SPAD pulser circuit (see $V_{EX}$ in Figure 1a), which can be externally adjusted from 2 up to 6.5 V, and the power supply for the analog and digital sections of the ASIC, which can be reduced to a single power supply ($V_{CC}$ in Figure 1a) with no significant drawbacks.

The final layout of a single block of the $64 \times 1$ fast gating AQC array, composed by eight AQCs, is shown in Figure 1b, with an overall length of 1.25 mm and 2 mm width: the eight SPAD pulser circuits with pads for wire bonding to the SPAD-dummy pair (see $S_i$ and $D_i$ in Figure 1b) are placed on the top side of the ASIC, while 20 pads for connection to the external circuitry (i.e., photon out signals, gate input signal, power supplies, external threshold voltage and one wire interface) are placed on the bottom side.

### 3.2. CMOS SPAD Array

The CMOS SPAD array was designed and fabricated in the same fabrication technology of the SPAD presented in Ref. [30]. The detector chip features 64 independent SPAD-dummy pairs laid out on a row. Each SPAD is a 50 µm square CMOS SPAD with rounded corners, and the pitch between adjacent SPADs is 75 µm, leading to a 66% overall fill factor. When the SPAD is biased 6 V above breakdown, the PDE is equal to 42% at 520 nm, with a DCR of 10 cps at 0 °C, which increases to a few

tens of kcps when operated at temperatures higher than 50 °C. Afterpulsing probability is lower than 1% when operated with 5 V excess bias and 50 ns hold off time. More data on SPAD performance are reported in [30].

The layout of the 64 × 1 SPAD array is shown in Figure 2a. The substrate and cathode contacts are shared among all the SPADs of the array, and bonding pads are placed on the chip side edges. Conversely, the independent contacts of SPAD anodes and dummy devices are placed on the top and bottom sides of the chip. It must be noted that the pitch between bonding pads of the SPAD-dummy pairs is double with respect to the SPAD pitch, and equal to the pitch of the bonding pads of the fast-gating AQC array. In order to exploit the full detector fill-factor, two different AQC arrays must be adopted, each one connected to one side of the CMOS SPAD array, as shown in Figure 2b, where the bonding scheme for the presented 16 × 1 fast-gated SPAD array is reported. Odd SPADs (i.e., 1, 3, . . . 15) are connected to the 8 × 1 AQC array on the bottom side of the SPAD array, while even SPADs (i.e., 2, 4, ... 16) to the 8 × 1 AQC array on the top side. In this first implementation the number of pixels was limited to 16, using a subset of the SPAD and fast-gating AQC arrays, since no commercially available TCSPC instruments provides more than eight input channels. Nevertheless, the high modularity of our design allows to easily scale up the detector up to a 64 × 1 fast-gated SPAD array.

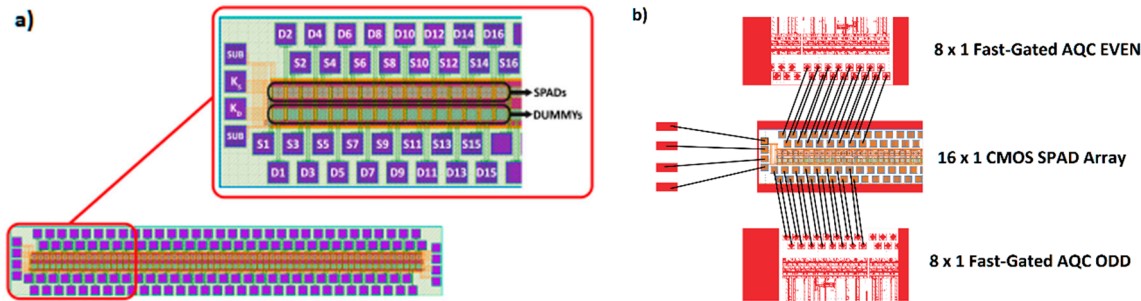

**Figure 2.** (**a**) Layout of the 64 × 1 SPAD array. SPAD and dummy contacts are placed at the top and bottom sides of the chip, whereas the common substrate and cathode contacts are placed on the left side. The inset shows a zoom of the 16 × 1 subset of the SPAD array adopted for the presented instrument. (**b**) Layout of the connections between the 16 × 1 Complementary Metal-Oxide-Semiconductor (CMOS) SPAD array and the two 8 × 1 fast-gating AQC arrays. The adoption of two different AQC arrays on both sides of the SPAD array allows the high fill-factor of the detector (~66%) to be exploited.

### 3.3. System Description

The presented 16 × 1 fast-gated SPAD array module is based on an assembly of two custom Printed Circuit Boards (PCBs):

- A chip-carrier board hosts the 16 × 1 SPAD array and the two 8 × 1 fast-gating AQC arrays. The three ASIC are wire bonded directly on the PCB.
- A control board hosts the chip-carrier board and its ancillary electronics, a microcontroller and a Complex Programmable Logic Device (CPLD) for system operation and timing signals.

A simplified block diagram of the developed instrument is shown in Figure 3: the chip-carrier board is connected to the control board for communication and power supplies, whereas the microcontroller guarantees a correct turn ON/OFF routine of the module and the CPLD is used to distribute the gate signal and the one wire interface to the two fast-gating AQC arrays. The instrument provides 16 independent outputs and requires only an input trigger signal for gate generation and a single 12 V DC power supply, with a power consumption ranging between 1.2 and 3.6 W, depending on gate repetition frequency and detector count rate, which did not impair the proper operation of the detector. Finally, a flange is mounted on top of the chip-carrier board allowing commercial optical filters and objective lenses to be mounted.

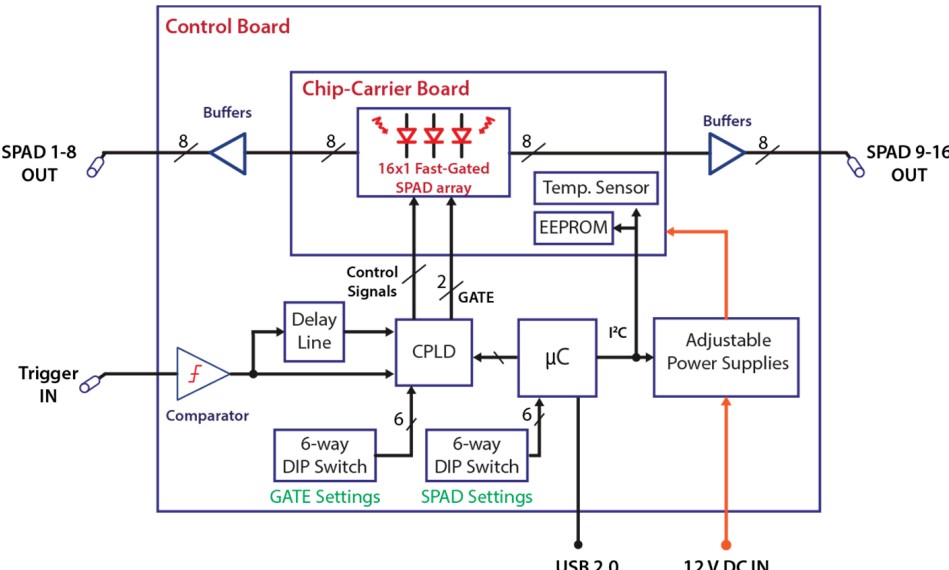

**Figure 3.** Simplified block diagram of the 16 × 1 fast-gated SPAD array module. The chip-carrier board, hosting the SPAD array and the fast-gating AQC arrays, is mounted onto the instrument control board, which features a Complex Programmable Logic Device (CPLD) for gate signal distribution and a one-wire interface for programming the AQC. A microcontroller is used to guarantee correct turn ON/OFF procedures and manages the adjustable power supplies for the chip carrier board via $I^2C$ bus. A trigger IN signal is required for time-gated operation and 16 independent output signals (see SPAD OUT in figure) are provided through 50 Ω coaxial cables. An USB 2.0 link is used for system debug purposes.

### 3.3.1. Chip-Carrier Board

The chip-carrier board hosts the SPAD array and the two AQC arrays, which are interconnected as shown in Figure 2b, and wire-bonded to the hosting PCB. A Negative Temperature Coefficient (NTC) resistor, mounted below the SPAD array (on the other side of the PCB) is used as temperature sensor to monitor SPAD operating temperature: its voltage drop is converted by an Analog-to-Digital Converter (ADC), whose output is read by the microcontroller through the module $I^2C$ bus. With respect to the module integration within a complete NLOS imaging instrument, in order to easily replace the chip-carrier board in case of damage, operating parameters of the SPAD array (breakdown voltage and excess bias voltage) and of the two AQC arrays (external thresholds and bias voltages) are stored in an EEPROM memory, and read by the microcontroller at system turn-on. Finally, the SPAD substrate voltage is derived from the SPAD cathode voltage power supply by means of a resistive trimmer, which allows the easy variation of the SPAD substrate voltage towards either the cathode or the anode voltage, and thus the modulation of the SPAD timing response in a trade-off between either narrower Gaussian peak (thus better SPTR) and longer exponential tail time constant, or broader Gaussian peak but shorter exponential tail time constant (for more details see [22] and [30]).

### 3.3.2. Control Board

Output pulses synchronous to photon detections are provided at the output of each AQC in CMOS logic levels. In order to provide output signals compatible with most TCSPC instrumentation, the 16 output signals of the fast-gated SPAD array are routed towards the control board, where 16 independent low-jitter output buffers convert output pulses into (Nuclear Instrumentation Module (NIM) logic levels (0 ÷ − 0.7 V) and drive the 16 output coaxial cables.

In order to provide a gate signal to the fast-gating AQC arrays, an external input is required (see Trigger IN in Figure 3). Through a 50 Ω impedance SMA connector, the Trigger IN signal is fed to the input of a wide-bandwidth any-level front-end comparator, the output of which is provided to

the CPLD, which allows for two different configurations for gate signal generation: (i) a synchronous copy of the Trigger IN signal is fed to the fast-gating AQC arrays, thus the duration of the input pulse determines the duration of the detector optical gate; (ii) it can be generated starting from the rising edge of the Trigger IN signal, as it triggers a monostable circuit designed within the CPLD and connected to an external delay-line, allowing to select the temporal duration of the optical gate between 25 and 60 ns at 5 ns steps. Nevertheless, such a resolution in gate-width selection (i.e., 5 ns) is not an issue in NLOS imaging application, as the main purpose of the fast-gated approach is the suppression of the strong ballistic light component. Additionally, as the instrument is supposed to be operated at 10 or 20 MHz repetition rate, as a good compromise between acquisition time and NLOS reconstruction volume, the adopted gate-width generally is respectively 50 ns and 25 ns (i.e., half measurement repetition period). The CPLD drives the gate inputs of the two fast-gating AQC arrays with the same gate signal. Nevertheless, since two different output ports of the CPLD are used, the architecture allows for different gating schemes by simply re-configuring the CPLD internal circuitry. Finally, a six-way DIP switch is used to allow the final user to select the configuration of the gate circuit allowing to (i) enable/disable the gate signal for the odd SPADs; (ii) enable/disable the gate signal for the even SPADs; (iii) select between external/internal gate generation; and (iv) select the temporal duration of the optical gate, when the internal gate generation circuit is used.

The control board microcontroller manages system operation, as it guarantees a safe power-up sequencing and controls the instrument turn-on routine, which consists in (i) downloading operating parameters from chip-carrier EEPROM; (ii) enabling chip-carrier power supplies in the correct order in order to avoid system failure; (iii) programing, through the CPLD, the two fast-gating AQC arrays; and (iv) finally enabling external threshold voltages for the AQC arrays. A six-way DIP switch is used to allow the end user to select: (i) threshold voltage for the Trigger IN front-end comparator among different values; (ii) the SPAD hold-off time as a multiple of the gate repetition period, selectable between 2 (i.e., the SPAD is re-enabled at the second valid gate after a photon detection) and 17 times the gate period (i.e., between 200 ns and 1.7 μs at 10 MHz repetition rate). Finally, an USB 2.0 link is used for the debug operation and for programming the chip-carrier EEPROM at the very first system turn-on, allowing the tuning and optimization of the SPAD and AQC array operating parameters.

Figure 4 shows a picture of the complete 16 × 1 fast-gated SPAD array module (dimensions are $12 \times 8 \times 5 \ cm^3$).

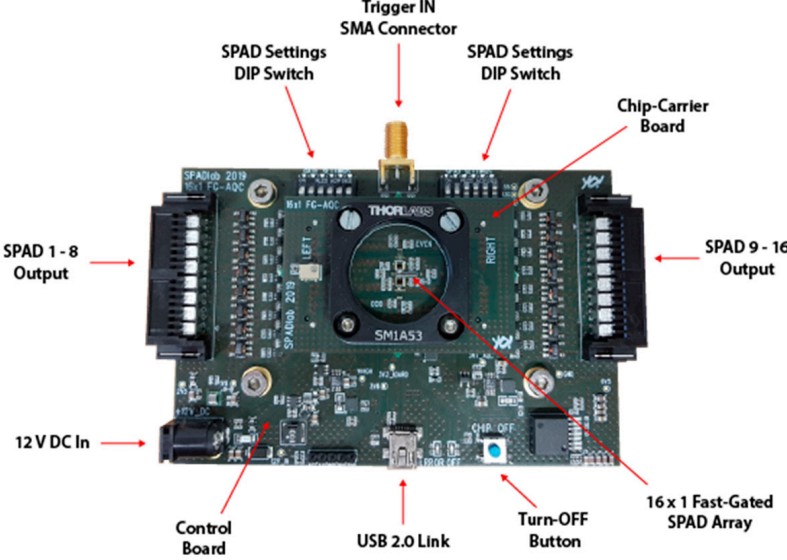

**Figure 4.** Picture of the complete 16 × 1 fast-gated SPAD array module. The chip carrier is equipped on the control board and a flange is used to allow for the mounting of optical filters and objective lenses.

## 4. Experimental Characterization

### 4.1. Fast-Gating AQC Array–Design Issues

During preliminary tests on the new AQC ASIC, we found a few issues in the integrated circuit, some of them with an impact on both quenching and timing performance of the system:

- Small size of the output buffer for the photon out signal (see $OUT_i$ pads in Figure 2b), leading to a weak driving current of the output pads, thus increasing the overall timing jitter of the detector. To limit its impact, the digital section of the ASIC is biased above 3.5 V in order to increase the output buffer driving capability (absolute maximum rating for this technology is 3.9 V) and reduce the overall timing jitter.

- Issue in the circuitry driving the degeneration MOS transistors in the SPAD pulser circuit (see $M_1$ and $M_2$ Figure 2a) resulting in having such transistors always OFF with the nominal 3.3 V power supply, leading to no effect of the reset BJT, and thus in a slow (longer than 1 ns) OFF-ON SPAD transition. A minimum 3.6 V power supply was found necessary in order to effectively overcome this limitation. The final power supply was set to 3.7 V for both the analog and digital sections of the ASIC, as this configuration proved best gating performance. All the measurement results provided in the following sections are obtained using a 3.7 V power supply.

- Too high resistance of metal rails for power supply of the SPAD pulser circuit ($V_{EX}$ in Figure 2a) and ground reference. The high resistance (estimated $> 1.5 \, \Omega$ for $V_{EX}$ rail) and the high peak current during SPAD transitions (estimated $> 1$ A) result in a strong electrical crosstalk within the AQC ASIC, affecting detector response linearity. Additionally, power consumption at the $V_{EX}$ power supply (higher that what expected after ASIC postlayout simulations) is proportional to the applied gate ON-time, maybe due to cross-conductance. This problem leads to high operating temperature for both the AQC arrays and the SPAD array, with an estimated SPAD operating temperature that may exceed 60°C, with non-negligible impact on detector DCR.

A redesign of the fast-gating AQC array ASIC is currently under way to correct the reported issues and improve detector performance.

### 4.2. Optical Crosstalk

During each avalanche, carriers are accelerated by the strong electric field across the depleted region and they can then loose energy also by emitting photons [31]. Such electroluminescence is spread across the infrared range and such emitted photons may eventually trigger an avalanche in adjacent SPADs, increasing detector noise and affecting detector temporal response. Optical crosstalk is a well-known flaw of multipixel SPAD detectors and can be ascribed to three different paths followed by secondary photons towards adjacent SPADs: (i) a Fresnel reflection at the interface between the silicon substrate and air; (ii) direct path toward the adjacent cells; and (iii) a Fresnel reflection between the air and the detector protective window (if present). This last contribution can be almost nulled by removing the detector protective window or by increasing its thickness, while the contribution through the direct path can be strongly mitigated by etching trenches between adjacent SPADs filled with opaque material [32]. Spurious counts due to optical reflections of secondary photons at silicon substrate interface are generally negligible with respect to the contribution through the direct path, but they make it impossible to completely suppress crosstalk effects even when contributions (i) and (ii) are removed [33].

In the presented design, the main contribution to optical crosstalk is due to photons propagating through the direct path, as in the adopted CMOS fabrication process no etch trenches can be introduced between adjacent SPADs. Two different measurements were carried out for evaluating the effect of optical crosstalk: in Figure 5, the system response to a 55 ps (FWHM) laser pulse at 850 nm and at 1 MHz repetition rate in three different conditions is reported. The SPAD array is biased 5 V above breakdown level (which is about 26.4 V at 60 °C) and driven with a 50 ns ON-time gate and 1 μs

hold-off time, with substrate biased close to anode voltage level, so as to reduce the long diffusion tail of the SPAD. All the 16 SPADs are active, driven by the same gate signal and flood illuminated by the laser beam. The blue curve reports the SPAD 3 response (adjacent to SPAD 2 and SPAD 4) whereas the SPAD 1 (adjacent only to SPAD 2) is reported on the red curve: a contribution to the laser pulse after the main Gaussian peak is visible in both curves, due to optical crosstalk between adjacent SPADs. By comparing the amplitude of the two curves at 3 ns, the contribution of the optical crosstalk to the acquired waveforms is double in SPAD 3 with respect to SPAD 1, in accordance to the number of adjacent SPADs to each detector. Different delays between SPAD 1 and SPAD 3 are due to variability of the propagation delay among different output buffers. To finally address the optical crosstalk as the cause of the waveform distortion, the green curve reports the SPAD 3 response when all the even SPADs (i.e., also SPAD 2 and 4) are OFF, where the timing distortion is nulled, as no photons are emitted by SPADs adjacent to SPAD 3.

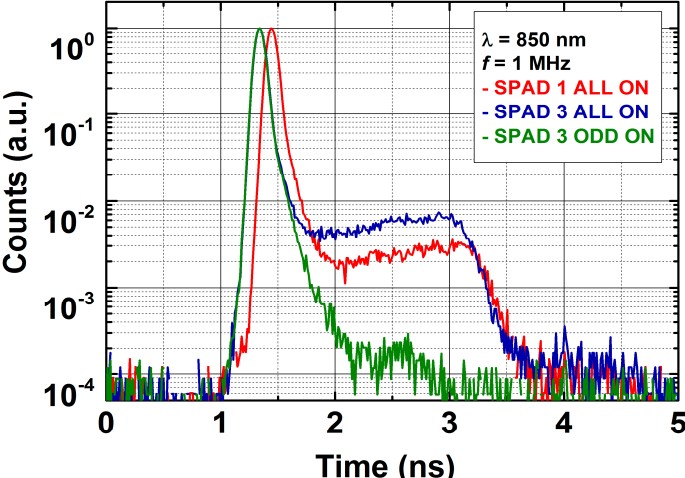

**Figure 5.** SPAD 1 (red line) and SPAD 3 (blue and green lines) responses to a laser pulse. By comparing SPAD 1 and SPAD 3 curves, it can be observed that crosstalk counts in SPAD 3 are almost double with respect to SPAD 1, as it receives contributions from both SPAD 2 and 4, whereas SPAD 1 only from SPAD 2. Optical crosstalk contribution is negligible on SPAD 3 when adjacent SPADs are turned OFF (green curve). Each curve is normalized to its main peak.

Figure 6 reports the SPAD 3 response to the same laser source when three different external references are employed for reading the avalanche pulses from even SPADs, i.e., those ones adjacent to SPAD 3. The external reference voltage, besides affecting the timing resolution of the system, influences the avalanche current quenching time, since a higher reference leads to a longer time required to detect the SPAD avalanche. Blue, red and green curves are acquired with progressively lower reference voltage from 2.45 V to 1.9 V: the crosstalk contribution to SPAD 3 response reduces by about 0.5 ns, thus further confirming that the waveform distortion is caused by optical crosstalk between adjacent SPADs.

A low threshold voltage allows the reduction of the duration of optical crosstalk contribution to about 1.5 ns. Further reduction of the external reference voltage is not possible as it leads to system instability. Optical crosstalk can be almost nulled by turning OFF half of the 16 SPADs (either even or odd ones), which leads to an 8 × 1 fast-gated SPAD array, with about 33% overall fill-factor. In any case, the contribution of spurious photons to the overall detection rate of each SPAD is just ~1%, thus it does not impair the adoption of the instrument for NLOS measurements.

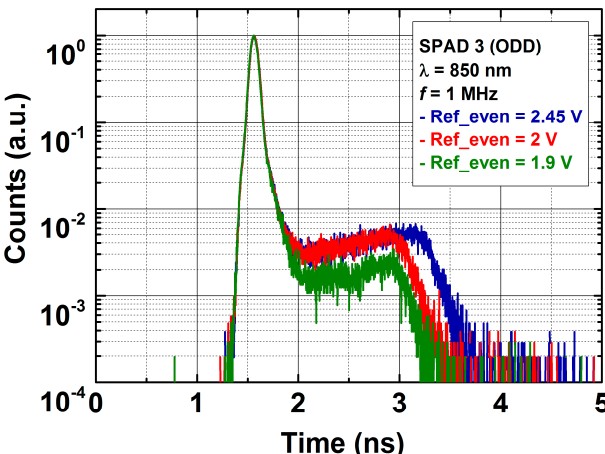

**Figure 6.** SPAD 3 response to a laser pulse at different external thresholds applied to even SPADs (adjacent to SPAD 3). By reducing the threshold voltage, the overall avalanche quenching time is reduced, leading to a shorter temporal duration of the optical crosstalk. Each curve is normalized to its main peak.

## 4.3. Electroluminescence

The long-lasting cross-talk contribution (about 1.5 ns) implies an avalanche current lasting at least as long as the optical crosstalk. Such a long quenching time is in contrast with postlayout ASIC simulation results and with previous AQC ASIC performance reported in [29], which exhibited a quenching time < 1 ns. To further investigate this issue, we measured the electroluminescence emitted by the SPAD under test by directly facing another SPAD detector in front of the 16 × 1 fast-gated SPAD array, in order to acquire photons emitted by the SPAD array during every avalanche [29].

Enabling all the 16 SPADs would result in higher background noise. Therefore, in order to effectively measure the quenching time of the avalanche current, only SPAD 1 is powered and enabled. As monitor detector we employed a commercial 20 µm SPAD module with 35 ps (FWHM) SPTR [34]. Inter-arrival times between SPAD 1 output pulses and the monitor detector were acquired for 24 h and the acquired electroluminescence curve of SPAD 1 is reported in Figure 7: the avalanche current within the SPAD driven by the AQC array lasts about 2 ns, in accordance with estimations obtained by means of the optical crosstalk measurements. The root causes of such a long quenching time inside the AQC ASIC are currently under investigation.

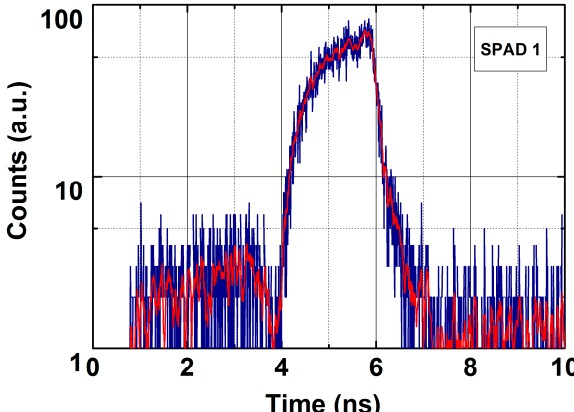

**Figure 7.** Electroluminescence of SPAD 1 avalanche acquired by directly facing a monitor SPAD detector in front of the 16 × 1 fast-gated SPAD array and acquiring inter arrival times between output pulses of the two detectors by means of a Time-Correlated Single-Photon Counting (TCSPC) instrument. Blue/Red line reports raw/filtered data.

### 4.4. Detection Linearity in Gated Operation

Figure 8 shows the photon detection uniformity in a 40 ns gate, obtained by acquiring only detector dark counts. A strong distortion preceding the gate falling-edge is observed, with an overall duration of about 5 ns. The cause of such distortion can be attributed to strong and fast current transients within the AQC array ASIC preceding the SPAD ON-to-OFF transition, as the distortion results being synchronous to the gate falling edge. The high power consumption from the $V_{EX}$ power supply rails, with peak current estimated being > 1 A during SPAD ON-to-OFF transition (see the P-MOS connected to SPAD anode node in Figure 1a) and the high resistivity of the AQC array ASIC metal rails, may result in both ground bounce phenomena and ringing effects on the $V_{EX}$ power supply within the ASIC, thus affecting detection linearity. Nevertheless, the amplitude of the distortion and its duration (i.e., 5 ns) does not impair timing performance of the instrument and thus its adoption for NLOS measurements.

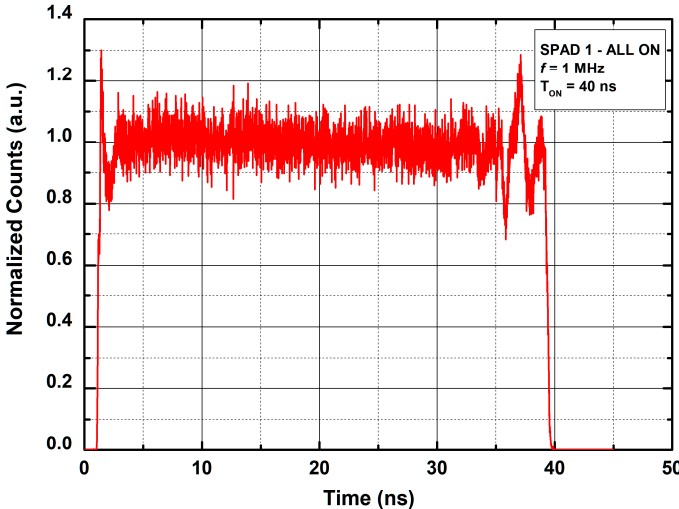

**Figure 8.** Final SPAD 1 photon counts distribution in a 40 ns ON-time gate at 1 MHz repetition rate when operating the SPAD array with 5 V excess bias voltage in dark conditions. The curve is normalized to its average value within the flat region of the optical gate window (i.e., from 5 to 30 ns).

As regards the rising edge of the optical gate, the TCSPC reconstruction shows a 20–80% rise time lower than 100 ps, much faster than design specifications and ASIC post-layout simulations. To better understand, we employed another technique for estimating the gate rise time: we acquired the detector response when illuminated by a laser pulse scanned over the gate rising-edge, as already in [29]. This preliminary measurement leaded to an OFF-to-ON transition time of about 500 ps (20–80%). The discrepancy between the two results can be due to the high threshold voltage applied to the AQC comparator, needed to prevent system instability. Avalanches ignited during SPAD OFF-to-ON transition experience a lower electric field with respect to avalanches ignited when the SPAD excess bias voltage is settled, thus having slower rising edge. High threshold voltage with low avalanche rising edge gives different delays from avalanche ignition to detection (slower initial avalanches are detected at a later time), eventually resulting in a faster OFF-to-ON transition in the reconstructed TCSPC waveform.

### 4.5. Single-Photon Timing Resolution

The detector SPTR was measured using an 850 nm laser source providing 55 ps (FWHM) optical pulses at 1 MHz measurement repetition rate (Advanced Laser Diode Systems). The SPAD array was biased 6 V above breakdown and enabled with 50 ns gates. Figure 9a reports the SPAD 1 response with substrate biased at a voltage close to the cathode voltage, so to have fast SPTR at the expense of a longer diffusion-tail time-constant. The same gate signal was applied to all the 16 SPADs. The acquired

waveform exhibits a FWHM equal to 69 ps, which accounts also for the laser pulse-width and the TCSPC electronics single-shot precision (i.e., 55 ps and 7 ps FWHM, respectively). By quadratically subtracting these values, the detector SPTR results equal 42 ps (FWHM). Results obtained in this operating condition are reported in Table 1 for all the 16 SPADs, with a SPTR ranging between 38 and 52 ps (FWHM).

To measure the diffusion tail time-constant with no distortion due to optical crosstalk, the SPAD 1 response was acquired in the same operating conditions mentioned above, but applying the gate signal only to odd SPADs (thus turning OFF the SPAD 2 adjacent to SPAD 1). When SPAD array substrate is biased close to cathode voltage (i.e., at 29 V, see red curve in Figure 9b), the diffusion tail time-constant results equal to 220 ps, whereas when substrate is biased close to anode voltage (i.e., at 3 V, blue curve) the time-constant is equal to just 55 ps, matching results obtained in [22].

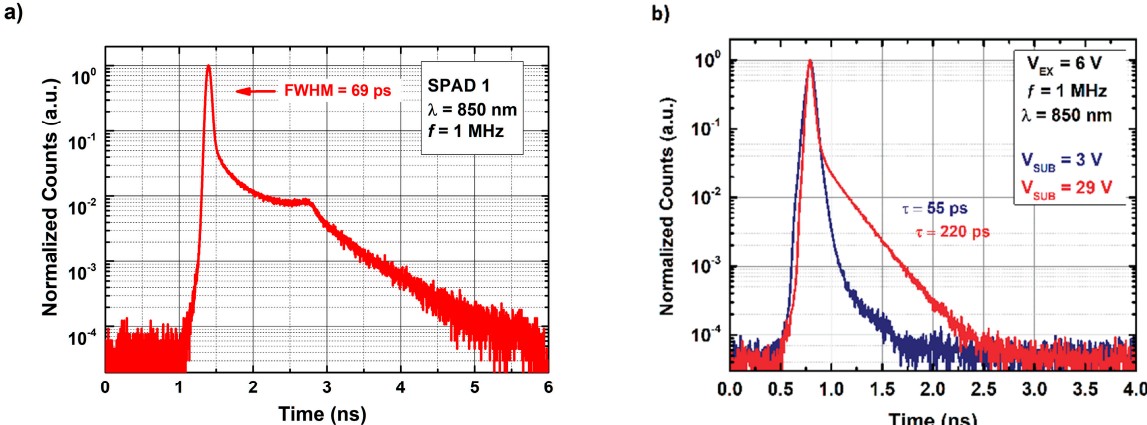

**Figure 9.** (**a**,**b**) SPAD 1 temporal response to a 55 ps (FWHM) laser pulse at 850 nm. SPADs are operated with 6 V excess bias voltage with 50 ns gate at 1 MHz. Each curve is normalized to its main peak. (**a**) All the 16 SPADs are enabled, with optical crosstalk component clearly visible after the main peak. (**b**) The same gate signal is applied only to odd SPADs and a comparison between different diffusion-tail time-constants at different substrate voltages is reported: a voltage close to anode one gives lower time constant (blue curve); a voltage close to cathode one gives enhanced Single-Photon Timing Resolution (SPTR) (red curve).

**Table 1.** Optical performance of the $16 \times 1$ fast-gated SPAD array when operated with 5 V excess bias.

| ODD SPAD # | SPTR [a] [ps] | DCR [b] [kcps] | EVEN SPAD # | SPTR [a] [ps] | DCR [b] [kcps] |
|---|---|---|---|---|---|
| 1 | 42 | 76 | 2 | 51 | 64 |
| 3 | 44 | 65.6 | 4 | 47 | 66.6 |
| 5 | 44 | 66 | 6 | 47 | 66.4 |
| 7 | 42 | 66 | 8 | 49 | 66.4 |
| 9 | 44 | 59.6 | 10 | 52 | 60 |
| 11 | 38 | 72 | 12 | 46 | 60.6 |
| 13 | 41 | 60 | 14 | 45 | 61 |
| 15 | 40 | 60.4 | 16 | 41 | 56.2 |

[a] Contribution of the laser source (55 ps–FWHM) is quadratically subtracted. [b] Dark counts acquired for 60 s and corrected for acquisition time, hold-off time and duty-cycle.

In order to test detector response at a higher pulse repetition rate, a 520 nm custom-pulsed laser source was used, providing laser pulses at 40 MHz repetition rate with 113 ps (FWHM) pulse-width. The SPAD array was operated 5 V above the breakdown with a 6 ns gate and substrate was biased close to anode voltage (i.e., 3 V). The acquired waveform is shown in Figure 10, with an overall FWHM of 130 ps.

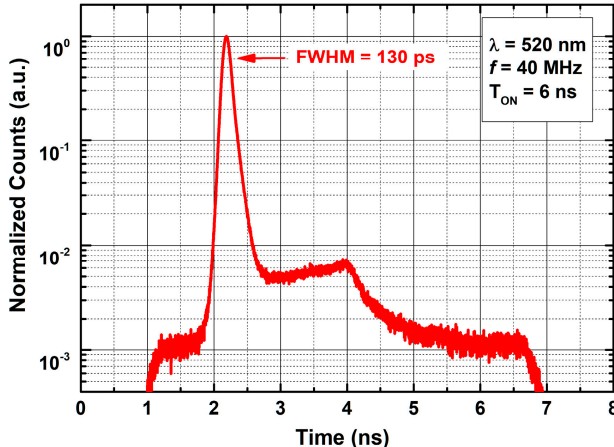

**Figure 10.** SPAD 1 response to a 113 ps (FWHM) at 520 nm when operated with a 6 ns gate at 40 MHz repetition rate. The curve is normalized to its main peak.

Finally, DCR of the 16 × 1 fast-gated SPAD array was evaluated by operating the detector at 10 MHz repetition rate with a 60 ns gate and with 1 µs hold-off time. Dark counts were acquired for 60 s and results were properly corrected for acquisition time, hold-off time and measurement duty cycle. Results are reported in Table 1. As mentioned, the high power consumption (about 1 W per AQC array in this operating condition) leads to a high operating temperature of the SPAD array, which may exceed 60°C, depending on ambient temperature and gate ON-time. This issue results in a DCR higher than 50 kcps, which still does not impair NLOS measurements since, generally, background noise due to LOS scene illumination is much higher than detector DCR.

## 5. NLOS Preliminary Measurement

The developed 16 × 1 fast-gated SPAD array has been installed in a complete NLOS imaging setup, which is based on: (i) a pulsed laser source, (ii) the 16 × 1 fast-gated SPAD array, (iii) a TCSPC module, and (iv) an object out of the detector LOS. A simplified representation of the measurement setup and photon paths through the scene is provided in Figure 11, whereas a picture of the measurement scene is shown in Figure 12: a 532 nm pulsed laser source emits picosecond optical pulses at 10 MHz repetition rate with 1 W average optical power (laser type 4, Katana HP amplified diode laser).

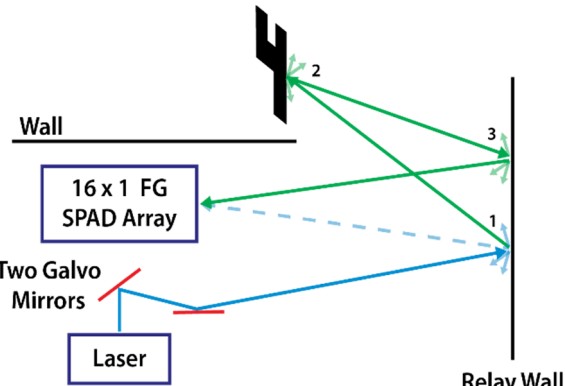

**Figure 11.** Photon paths through the scene of the NLOS experimental setup. Two galvo mirrors are used to project the laser beam on the relay wall, from which photons are scattered through the scene (first bounce). While ballistic photons reaching the 16 × 1 fast-gated SPAD detector are time-filtered, some of the remaining photons will reach the hidden object and will be reflected back to the relay wall and, finally, to the detector.

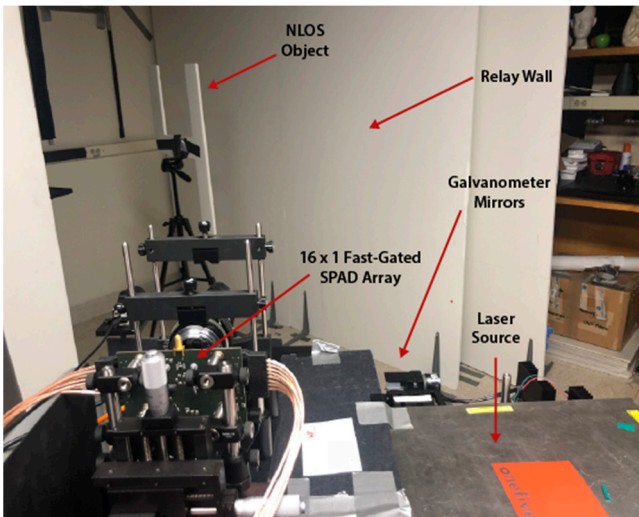

**Figure 12.** Picture of the Non-Line-Of-Sight (NLOS) measurement scene, including: the $16 \times 1$ fast-gated SPAD array, the laser source (behind the grey screen) and the galvo mirrors, the relay wall and finally the NLOS object (a number "4" made of white paper).

Two galvo mirrors project the laser beam on the relay wall (about 2 m from the imaging system), where a squared area with a $150 \times 150$ cm grid, with 1 cm spacing between points (i.e., $1.5 \times 1.5$ m$^2$), is scanned. Some photons are reflected off the relay wall towards a hidden object, i.e., a number "4" made of white paper (dimensions $40 \times 60$ cm$^2$, about 1 m far from the relay wall), and back-reflected photons may eventually reach the $16 \times 1$ fast-gated SPAD array module, which is focused on a 1 cm$^2$ area on the relay wall by means of an objective lens mounted onto the chip carrier flange. The detector is operated with a 60 ns gate by means of the internal gate generation circuit, applied only to even SPADs, thus using the detector in an $8 \times 1$ configuration. The overall acquisition time is equal to 8 min for the whole grid, thus the exposure time per point is equal to only 21.3 ms/point, i.e., 8 min/($150 \times 150$). Finally, an eight-channel TCPSC instrument (Picoquant Hydraharp [35]) acquires the optical waveforms with 12 ps (RMS) single-shot precision.

The use of such an intense laser source allows the reduction of the overall measurement time to few minutes for the whole reconstruction, nevertheless the adoption of a fast gating-approach (with a rising edge faster than 1 ns) is mandatory to reject the strong ballistic contribution of light reflected by the relay wall, thus allowing the detection of only nonballistic photons.

*Reconstruction of the Hidden Object*

The NLOS reconstruction technique here adopted is a version of the back projection algorithm described in [18]. Although this technique provides worse resolution and quality of the reconstruction with respect to other approaches (e.g., convex optimization algorithm [16]), the adopted method requires less data, does not rely on any assumption about the geometry of the hidden scene and allows discrimination between hardware and software induced artifacts.

For every time bin of each acquisition, this algorithm takes into account (i) the number of counts within the time-bin N; (ii) the corresponding photon arrival time t; (iii) the coordinates of the spot on the relay wall observed by the detector ($x_0$, $y_0$); and (iv) the coordinates of the laser spot on the relay wall ($x_i$, $y_i$), to determine shape and position of objects in the hidden scene. For every photon count—N(t, $x_0$, $y_0$, $x_i$, $y_i$)—is projected from the native five dimensional space in ellipsoid in the three dimensional Cartesian space V(x, y, z), with voxel dimensions adjusted according to the timing resolution of the NLOS imaging system. The backprojection results in a confidence map that describes the likelihood of the light being reflected by the different voxels in the reconstruction volume [7]. The thickness of the back projected ellipsoids depends on the timing resolution of the system, on the laser spot size

and on dimensions of the area observed by the SPAD. The obtained results are reported in Figure 13, where a picture of the NLOS object is reported for seven SPADs (odd SPADs from 1 to 13), proving the suitability of the presented instrument in NLOS applications. SPAD 15 results are not reported due to the limited number of seven input channels of the TCSPC instruments, when using the eighth input for measurement synchronization.

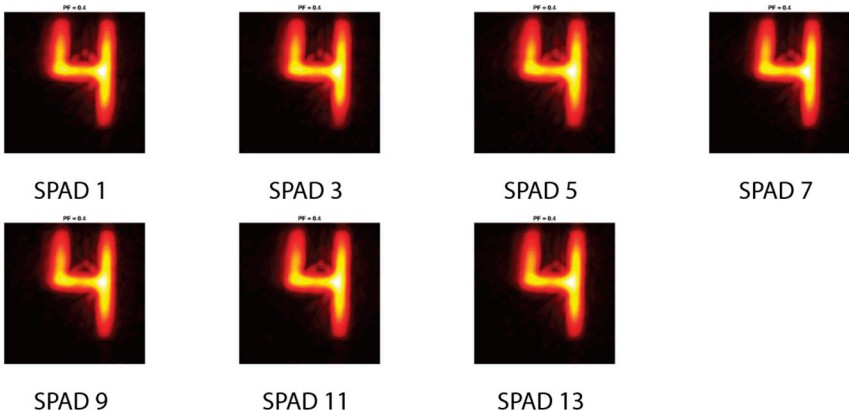

**Figure 13.** NLOS reconstruction of the hidden object, a number "4" made of white paper (dimensions are $40 \times 60$ cm$^2$), as seen from the relay wall. Images where acquired by enabling only odd SPADs and overall measurement acquisition time is equal to 8 minutes and we acquired about $4 \cdot 10^7$ photons/pixel. Each reconstruction takes approximately 6 minutes.

## 6. Conclusions

In this paper, we presented a novel single-photon detector based on a $16 \times 1$ CMOS SPAD array operated in fast-gated mode, designed for NLOS imaging applications within the framework of the DARPA REVEAL program. The instrument proved a SPTR resolution better than 50 ps (FWHM) with sub-ns SPAD OFF-to-ON transitions up to 40 MHz repetition rate. The detector has been successfully adopted in a NLOS imaging system, proving its suitability for reconstructing a hidden object. Additionally, the adoption of a multipoint detection, together with novel NLOS reconstruction techniques, may lead to an overall NLOS imaging acquisition time of just a few seconds, allowing for detection of moving hidden objects in complex scenarios. The possibility to use non-confocal acquisition together with the phasor-field method allows the overcoming of current limitations of most state-of-the-art NLOS imaging systems, allowing for multipixel acquisition and still guaranteeing optimal performance in reconstructing complex NLOS scenes.

The modular structure of the presented detector allows to scale the array dimensions up to a $64 \times 1$ fast-gated SPAD array, to be paired with a multichannel TCSPC system based on FPGA. Design issues in the fast-gating AQC array currently prevent the possibility to scale up the detector due to excessive heating, but a custom optical package and detector cooling could allow to overcome this limitation and reach the full instrument capability. Future designs may also take advantage of the possibility to operate an InGaAs/InP SPAD array with the same fast-gating AQC array ASIC, thus allowing the adoption of near-infrared pulsed laser sources for building eye-safe NLOS imaging instruments, with improved rejection of background noise.

**Author Contributions:** Conceptualization, M.R. and A.T.; methodology, M.R., M.B., F.V., A.T., A.V.; software, M.R. and J.H.N.; validation, M.R., J.H.N.; investigation, M.R. and J.H.N.; data curation, M.R and J.H.N.; writing—original draft preparation, M.R.; writing—review and editing, A.T.; visualization, M.R.; supervision, A.T. and A.V.; funding acquisition, A.V. and A.T. All authors have read and agreed to the published version of the manuscript.

**Funding:** This work was funded by DARPA through the DARPA REVEAL project (HR0011-16-C-0025).

**Conflicts of Interest:** The authors declare no conflict of interest.

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
