# Peer review of "Fast-Gated 16 × 1 SPAD Array for Non-Line-of-Sight Imaging Applications"

_instruments, doi:10.3390/instruments4020014_

Round 1

Reviewer 1 Report

The paper is well written and the technical content is from interest for the reader. I can recommend the paper for publication but I would like to provide some remarks in order to improve the quality of the paper:

1.) In the introduction you write "...in acquiring the LOS image, NLOS techniques rely..." but I guass you mean "...in acquiring 36 the LOS image. NLOS techniques rely..."

2.) The subtitel 2.1. should be removed or another subtitel should be inserted. However, you need at least 2.1 and 2.2 otherwise a subtitel makes no sense. The same for 3.1.1.

3.) Please give the y-achsis is Figure 9b a name.

4.) Please specify the term "normalized counts" in your main text.

5.) Why is Figure 6 not normalized?

Reviewer 2 Report

The paper describes a system for NLOS measurements, based on a linear array of SPADs. The system is based on two types of chips, one for the SPAD detectors, one (AQC chip) containing a circuit that is supposed to turn-on SPADs in a very fast way and in general provide the time-gating functionality.

Algorithms (published elsewhere) are also included in the system.

The paper is very well written and pleasant to read. Nevertheless, the system lacks of novelty. The AQC chip is very similar to one published before. The SPAD chip is very simple. In terms of integration, there are already several fully integrated linear and 2D implementations of SPAD arrays; some of them integrate fast SPAD charging mechanisms, too. On top of that, I can't see the need for a 2-chip implementation if the SPAD array is made in a CMOS standard technology! If I'm missing something, probably other readers will miss it too, so please highlight it in the text.

Apart from that, what is missing is a comparison of the suggested method with respect to other hardware approaches. For example, why is the proposed 2-chip, 1D-array configuration better than a SPAD imager like the one in [21]?
In particular, why should gating provide better performance than timestamping?

Other specific comments are provided below:

88 - How can a SPAD compete with I-CCDs in terms of PDE?
95 - Why third? How about first and second bounce?

110-116 - "The reconstruction method exploited in [7] is based on a filtered model presented in [18], tailored to reduce the computational resources required to perform the reconstruction, but computation times of few hours are required due to the large amount of data and the complex 112 algorithms to be employed for the reconstruction [25]. Such work reduces the computational resources (up to three orders of magnitude) required to perform the reconstruction as it decreases the overall number of computation by considering only the data resulting from the intersects of 115 several ellipsoids whose poles are the light source and the detector [25]."

156 - Which "dummy side"? The paper should be self-consistent. A dummy structure is introduced in the abstract, but no explanation or description is given.
160 - The same holds for the "16 bit LFSR counter"

169 - The final 64 × 1 AQC => The "final 64x 1 AQC", which the authors refer to, has never been introduced before and it is not clear what it is. Isn't the SPAD array made of 16 elements?

186 - "64 independent SPAD-dummy pairs" => I'm confused. The abstract talks about a 16 x 1 array; the introduction about a system that "features 16 independent SPADs with the possibility to increase the number of detectors up to 64 pixels". What is the final number? At this point of the paper, it seems that you are talking about a system which is made assembling together multiple samples of two types of chips. 1 type is devoted to quenching, gating and front-end, and it has 8 channels. The 2nd type is an 16 x 1 array of SPADs with a dummy structure. So to achieve 64 channels you need 8 samples of the first type and 4 samples of the second type.
In any case (being what I wrote correct or not), I suggest to describe the system more explicitly, possibly with the aid of a figure (either a picture or a block diagram).

280-285 - "A minimum 3.6 V power supply was found necessary in order to effectively overcome this limitation. The final power supply was set to 3.7 V" => what's the fast-gating time at these values?

316 - Does it mean that the avalanche is sustained for >1.5 ns? This seems a long time!!! I don't understand the need for an external quenching then... A simple passive quenching mechanism implemented close to the SPAD would provide faster quenching.

358-361 - I'm confused. Do I understand correctly that you measure the same thing (uniformity across gate time) in two ways, the first using DCR (from an arbitrary START to 1st SPAD triggering due to a dark) and the second using a laser (using increasing delays for the laser trigger)?
The two measurements provide different results; how do you explain this?

398 - Distances are not shown. Please add them

410 - "Nevertheless the adoption of a fast gating-approach is 410 mandatory to reject the strong ballistic contribution of light reflected by the relay wall"
From previous characterization, it seems that the chip does not provide the expected speed in turning on the SPAD. This sentence somehow contradicts the need for a fast-gating.
I think that a sentence that explains how much you lose not having the expected performance (or you would gain if you had the expected performance) is needed.

414-418 - How about processing time?

Reviewer 3 Report

The paper presents a single-photon detector designed around technologies which are well-known to the authors, but specifically tailored to Non-Line-Of-Sight (NLOS) imaging applications. The detector has been characterised in detail and its pros and cons presented. What would add value to the paper is a) a better description of the novel aspects (over and beyond the combination of known building blocks), and b) a better comparison with other NLOS approaches, so as to improve the reader’s understanding of where this work fits in the global NLOS landscape.

It would also help the better address how to overcome the sometimes important limitations indicated at page 8, in particular when scaling up the detector.

Concerning the state-of-the-art, it might be necessary to add references to recent works of the Laurenzis, Faccio and/or Wetzstein groups, amongst others.

NB: some images appear to be downsampled and of lower quality. It is not immediately evident that the 16 SPADs are part of a larger 64 SPAD array.

DETAILED COMMENTS:

Abstracts: some additional (quantitative) details would help.

Page 2: an overall schematic NLOS image would help already at this point. Maybe also a definition of DR, SNR, for photon-counting sensors.

Page 2-3: comparison with other NLOS systems is mostly qualitative. A summary table with the main specifications would certainly help the reader.

Page 3:

- The relationship between the sentence at lines 110-113 and the following one is not clear.

- l. 122-123 “an acquisition time of several minutes”: real-time NLOS - at least tracking – seems to have been already demonstrated by Faccio and/or Wetzstein.

Page 4:

- Please specify in which process the designs have been carried out.

- The transistor sizes are likely to be in nm, but it might help to specify it.

- Fig. 1 (b): please add size.

- The InGaAs/InP SPAD reference might be confusing to the reader. I suggest to briefly clarify this aspect.

Note: it is likely necessary to provide a short summary of the previous fast-gating AQC, otherwise the discussion becomes too hard to follow.

- Several specifications seem to be quoted at temperatures which are different from the final one, due to the system heating. This can lead to some confusion and should be clarified.

Page 5: quite high power consumption (1.2-3.6 W) -> is there a power breakdown? What are the consequences, at system and reconstruction level (e.g. due to the high DCR)? Is any cooling possible?

Page 9: optical crosstalk: was a similar effect seen in other chips implemented in the same technology? Also, you might want to add a comment to sentence 328-329 – the reader might not have realised that he/she is looking at a semi-log plot, and that the optical crosstalk is therefore modest, at least for this application.

Page 11, l. 349: “with respect to average in-gate value and 5 ns duration” – the 5 ns reference is not entirely clear. Similarly, lines 358-361: why are 100 ps mentioned initially, then 500 ps?

Page 13: 1 W average optical power is very high. The authors might want to comment on this from the safety/practicality perspective. Btw, which is the laser type?

Page 14: Figure 13: does it make sense to combine all SPAD inputs and reconstruct a single image (in less time)?

Page 15 Conclusions: a better comparison with the SoA would be helpful, whether single point, line array, or 2D SPAD-based configurations. Also, it’s not entirely clear how to overcome the sometimes important limitations indicated at page 8, in particular when scaling up the detector.

__________

Typos/wording: see highlighted text in enclosed PDF.

__________

Reviewer 4 Report

Using actively quenched SPADs for non-line-of-sight detection is a new possible application for single photon detectors.  The authors have presented preliminary but promising results for this novel application.

The work was clearly described, and the data were carefully taken.  I recommend publication of the paper with some minor modifications:

(1) There have been detailed discussions on the effect of optical crosstalk due to electroluminescence.  It would be nice to discuss possible methods to minimize this effect.

(2) The afterpulsing effect should be discussed in more detail.  

(3) The device shows 3-4 orders of magnitude increase in dark count when the temperature changes from 0C to 40C.  But there is no discussion about the device temperature durng operation.  From the dark count rate of around 60KCPS, does the device operate at 40-50C.  How does the increase in the DCR affect the performance?

(4) What is the photon arrival rate in the test setup? 

Round 2

Reviewer 2 Report

All aspects have been addressed.

Congratulations to the authors!

Reviewer 3 Report

- In general the authors replied to all comments. The English of some newly added parts should be rechecked.

- I don't manage to see the following addition (in yellow) in the revised text:

-----

  1. Page 15 Conclusions: a better comparison with the SoA would be helpful, whether single point, line array, or 2D SPAD-based configurations. Also, it’s not entirely clear how to overcome the sometimes important limitations indicated at page 8, in particular when scaling up the detector.

OK, we modified the text including a better comparison with state-of-the-art solutions for NLOS.

We are confident that the main chip limitations reported at page 8 (i.e. small size of the output buffer for the photon out signal, issue in the circuitry driving the degeneration MOS transistors in the SPAD pulser circuit, too high resistance of metal rails for power supply of the SPAD pulser circuit) can all be solved with a proper redesign of the chip in either the same or similar technology.

-----

- I am not sure that the following addition to another reviewer's comment is correct (I understand that ICCD cameras rely on photocathodes, whose quantum efficiency is not very high), and would therefore suggest that the authors verify carefully their correction:

Other specific comments are provided below:

  1. 88 - How can a SPAD compete with I-CCDs in terms of PDE?

Yes, we agree that I-CCDs have far better PDE. We modified the text to clarify that there is no comparison between the PDE of SPADs and I-CCDs:

“…by SPAD detectors: despite a lower Photon Detection Efficiency (PDE) compared to I-CCDs, they guarantee…”.